# Survival assessment and pre-diagnostic risk factors for lung cancer incidence: Insights from the Golestan Cohort Study

Seyedeh Fatemeh Mousavi[1☉], Sahar Masoudi[1☉], Negar Rezaei[2], Farzad Pourghazi[1], Maryam Sharafkhah[1], Maysa Eslami[1], Akram Pourshams[2], Hossein Poustchi[3], Gholamreza Roshandel [4], Rasoul Aliannejad[5,6‡], Sadaf G. Sepanlou [1‡], Reza Malekzadeh[2]

**1** Digestive Diseases Research Center, Digestive Diseases Research Institute, Tehran University of Medical Sciences, Tehran, Iran, **2** Digestive Oncology Research Center, Digestive Diseases Research Institute, Tehran University of Medical Sciences, Tehran, Iran, **3** Liver and Pancreatobiliary Diseases Research Center, Digestive Diseases Research Institute, Tehran University of Medical Sciences, Tehran, Iran, **4** Golestan Research Center of Gastroenterology and Hepatology, Golestan University of Medical Sciences, Gorgan, Iran, **5** Advanced Thoracic Research Center, Tehran University of Medical Sciences, Tehran, Iran, **6** Division of Pulmonary and Critical Care, Shariati Hospital, Tehran University of Medical Sciences, Tehran, Iran

☉ These authors contributed equally to this work.
‡ RA and SGS also contributed equally to this work
* sepanlou@yahoo.com, sgsepanlou@tums.ac.ir

## Abstract

### Background

Lung cancer remains a pressing health issue globally. This study investigates survival rates and the impact of pre-diagnostic factors on lung cancer incidence in Golestan Cohort Study (GCS).

### Method

The GCS, initiated in 2004 with enrollment concluding in 2008, comprises 49,783 individuals aged 40–75 from the Golestan province in northeastern Iran. Our analysis included all cases of lung, tracheal, and bronchial cancers diagnosed under ICD-10 codes C33-C34 from the study's inception to 2022, tracking participants until death. A sensitivity analysis, excluding lung cancer cases diagnosed within the initial 24 months of follow-up, was performed to address the reverse causation bias from previously undiagnosed conditions at baseline.

### Results

Out of 49,783 participants in the study, 132 were diagnosed with lung cancer, of whom 130 died by the end of the study. The age and sex-standardized incidence rate stood at 20.39 per 100,000 person-years. The median survival post-diagnosis was approximately four months, with one-year and five-year survival rates at 18.67% and

**Data availability statement:** Data are available from the Digestive Diseases Research Institute, Tehran University of Medical Sciences, Tehran, Iran. Address: North Kargar Street, Shariati Hospital, Digestive Diseases Research Institute. Phone: 82415000 Fax: 82415400 Email: shariati.ddri@gmail.com Postal Code: 1411713135.

**Funding:** The author(s) received no specific funding for this work.

**Competing interests:** The authors have declared that no competing interests exist.

1.56%, respectively. Sensitivity analyses identified advanced age, male sex, opiate use history, pack-years of cigarette smoking, and the utilization of non-gaseous energy sources as lung cancer risk factors. In contrast, high physical activity and a BMI of 25 or higher were inversely associated with lung cancer risk.

## Conclusion

Our study highlights the critical burden and low survival rates of lung cancer in resource-limited regions. Mitigating key risk factors and enhancing access to diagnostic and treatment services through targeted public health policies and comprehensive strategies are essential for ensuring equitable healthcare and improving lung cancer outcomes for underserved populations.

## Introduction

Cancer continues to be a significant public health challenge globally, accounting for one-sixth of all deaths worldwide [1]. Lung cancer, in particular, is the leading cause of cancer-related mortality worldwide for both men and women [2]. The burden of lung cancer extends beyond mortality; in 2019, it accounted for approximately 45.9 million disability-adjusted life years (DALYs) lost, underscoring the disease's significant economic and health ramifications [3]. Unfortunately, the incidence of tracheal, bronchial, and lung cancer doubled from 1990 to 2019, reflecting an increase of approximately 1.1 million new cases [3]

Lifestyle and environmental factors are recognized as contributors to lung cancer, with smoking being the preeminent risk factor [3–5]. Smoking reduction policies have yielded measurable improvements in lung cancer incidence rates and associated mortality [6, 7]. Identifying additional potential risk factors is crucial, as it informs policy measures to reduce this disease's burden. Early detection of lung cancer is also an important step toward reducing mortality [8]. Evidence supports the effectiveness of screening programs designed for high-risk populations. The success of such screening initiatives relies on accurately identifying and targeting populations at increased risk, thereby improving program outcomes [9, 10].

Although significant strides have been made in understanding lung cancer's incidence, survival, and risk factors, the current body of research is predominantly focused on developed countries. This skew has resulted in a knowledge gap in understanding the survival and risk factors pertinent to lung cancer in low-resource settings. The diverse demographic profiles and exposure risks found in developing countries offer a rich yet underexplored context for cancer prevention and control strategies [11–13]. Furthermore, inconsistencies in the literature regarding risk and protective factors for lung cancer highlight the need for more detailed prospective studies to reconcile these differences [14].

This prospective cohort study addresses a crucial void in cancer epidemiology by exploring the incidence, survival, and pre-diagnostic factors of lung cancer within

a developing region where incidence rates are rising [15]. By delineating these relationships, we intend to improve the scope and efficacy of prevention measures for lung cancer.

## Materials and methods

### Design and study population

The Golestan Cohort Study (GCS) is a population-based prospective cohort study initiated in January 1, 2004 to investigate the etiology of upper gastrointestinal cancers, specifically, esophageal cancer. It included individuals aged 40 –75 from the Golestan region. Recruitment was accomplished on June 30, 2008. Specifically, 20% of the participants were from an urban area, Gonbad City, and the rest were recruited from all 326 villages in Aq-Qala, Kalaleh, and Gonbad counties in the province of Golestan. By June 2008, the recruitment target of roughly 50,000 individuals was met. After implementing the pre-established criteria, the final cohort size amounted to 49,783 participants. Exclusion criteria were: unwillingness to participate at any stage of the study for any reason; being a temporary resident; and having a current or previous diagnosis of an upper gastrointestinal cancer. The GCS's comprehensive design and methodology have been elaborated on in previous publications [16].

For this study, data were accessed on December 31, 2022. We included all participants diagnosed with lung, tracheal, or bronchial cancers diagnosed under International Classification of Diseases, Tenth Revision (ICD-10) codes C33-C34 from the beginning of the study and monitored them until their death. To counter potential bias from reverse causation— stemming from previously undiagnosed lung cancer—we conducted a sensitivity analysis that excluded cases detected within the first two years of follow-up.

### Exposure assessment

Trained general physicians and nutritionists conducted interviews and performed concise physical examinations. A comprehensive questionnaire was administered to capture vital demographic, socioeconomic, medical, behavioral, and lifestyle characteristics. This questionnaire included inquiries about sex, age, place of residence, ethnicity, education level, opium and tobacco use, socioeconomic status, fuels used for heating and cooking, contact with animals, and physical activity levels. Importantly, before the actual enrollment of participants, a preliminary pilot study was carried out to evaluate the feasibility of the prospective cohort study. This initial phase was crucial for assessing participant response rates, refining methods for gathering nutritional and lifestyle data, and establishing protocols for tracking health outcomes. To further assess the data's repeatability in the actual cohort, the entire enrollment process was repeated with a subset of participants from rural areas at a mean interval of 45 months. The findings demonstrated excellent consistency between the initial and subsequent data collections [16].

Regarding smoking habits, participants were asked about their smoking history to calculate the cumulative consumption in pack-years, assuming 20 cigarettes per pack. For opium consumption, participants' usage history and routes of administration were documented. In a previous study, the accuracy of self-reported opium use in questionnaire data was confirmed through a high correlation with urine test results for codeine or morphine. Additionally, there was strong agreement between self-reported tobacco smoking, or nass use and urinary cotinine levels, showcasing the reliability and precision of the substance use data collected [16, 17]. Body Mass Index (BMI) was calculated by dividing weight in kilograms by height in meters squared, with WHO-defined cut-offs categorizing individuals accordingly. A BMI below 18.50 kg/m² was classified as underweight, 18.50–24.99 kg/m² as normal, 25–29.99 kg/m² as overweight, and a BMI of 30 kg/m² or higher as obese. Physical activity levels of the participants were categorized based on their median Metabolic Equivalents (MET) scores. Participants with scores above the median were deemed active, while those with scores below the median were deemed inactive. Regarding the source of energy usage, various fuels, including natural gas, kerosene, diesel, and biomass, are utilized for cooking and household heating in Golestan. For assessing socioeconomic status, participants

were categorized into quartiles based on a combined wealth score. This score was derived from multiple correspondence analyses of household assets, with the methodology detailed in prior publications [16,18].

### Follow-up and outcome assessment

The Atrak Clinic databases and the Golestan Cancer Registry are systematically reviewed each month to detect cancer cases within the cohort. Furthermore, at the outset of the study, participants were instructed by the GCS team to pro-actively inform the research staff of significant health events, including hospital admissions or diagnoses of new major illnesses. The team diligently records and confirms these events. Additionally, the GCS performs annual follow-ups with participants to assess their ongoing health conditions. During these follow-ups, which are carried out either by phone calls or home visits, the team employs a comprehensive questionnaire to capture any changes in the participants' health, including new diseases or hospitalizations since the previous contact. For validation purposes, two independent external internists review all pertinent clinical documents to assign ICD-10 codes and to determine the onset dates for each disease case. Any discrepancies between the coding by these internists are referred to a third, senior internist for resolution, which ensures the precision of the final diagnosis. In cases where valid medical records are not available, prior research has validated the verbal autopsy method for its high reliability in determining causes of death [19].

### Statistical analysis

In this study, we present qualitative data using frequencies and percentages, while quantitative data were summarized with means and standard deviations. We calculated cancer incidence rates as new cases per 100,000 person-years in the population at risk. Median survival, the time until half of the participants had passed away, was expressed in months. Survival analysis was performed using the Kaplan-Meier method.

Univariate and multiple Cox regression models were employed for the analytical approach. Variables with a p-value less than 0.2 in crude analysis were considered for the multiple regression model [20]. The final multiple model established statistical significance at $p < 0.05$. Data analyses were performed using Stata software (version 12).

### Ethics approval statement

The GCS received ethical approval from the Digestive Disease Research Institute of the Tehran University of Medical Science (Ref: FWA00001331), the U.S. National Cancer Institute (NCI), and the International Agency for Research on Cancer (IARC) (Ref: CN/23/3). All participants were informed about the study's scope and provided written consent before participating.

## Results

Out of the 49,783 individuals enrolled in the study, 132 were diagnosed with lung cancer, with females representing 29.6% and males representing 70.5% of these cases. Among the total cohort, a history of smoking was significantly more common in males, with 38.6% reporting smoking habits, as opposed to only 1.6% of females, highlighting a marked sex disparity in smoking prevalence.

The age and sex-adjusted incidence rate of lung cancer was 18.78 per 100,000 person-years, with a rate of 32.19 for males and 9.42 for females. Following a diagnosis of lung cancer, the median survival time was approximately four months. Survival rates at one year and five years post-diagnosis were 18.7% and 1.6%, respectively.

Tables 1 and 2 outline the relationship between specific pre-diagnostic factors and lung cancer risk, as established by univariate and multiple analyses, respectively. The univariate analysis indicates that risk factors such as male sex, advancing age, history of smoking, opium consumption, and the use of non-gas fuels are associated with lung cancer risk. In contrast, higher levels of physical activity and a BMI of 25 or above correlate with a diminished risk.

**Table 1. Demographic characteristics and univariate analysis of pre-diagnostic factors impacting lung cancer incidence.**

| | Incidence/ person-year | Incidence rate (per 100,000) | Hazard Ratio | P value |
|---|---|---|---|---|
| **Sex** | | | | |
| Female | 39/413808.76 | 9.42 | ref | |
| Male | 93/ 288926.54 | 32.19 | 3.41 (2.35-4.96) | **0.000** |
| Total | 132/ 702735.3 | 18.78 | | |
| **Age at diagnosis*** | | | | |
| 35-45 | 0/235.34 | 0 | | |
| 45-55 | 26/4285.27 | 606.72956 | | |
| 55-65 | 47/8173.7 | 575.01499 | 1.06 (1.044-1.08)* | **0.000** |
| 65-75 | 40/6915.65 | 578.39827 | | |
| ≥75 | 19/3555.48 | 534.38636 | | |
| **Ethnicity** | | | | |
| Turkmen | 98/528989.21 | 18.52 | ref | |
| Non-Turkmen | 34/173746.09 | 19.57 | 1.04 (0.71-1.54) | 0.829 |
| **Education** | | | | |
| Illiterate | 88/ 485807.5 | 18.11 | ref | |
| Educated | 44/ 216927.8 | 20.28 | 1.13 (0.78-1.62) | 0.521 |
| **Residence** | | | | |
| Rural | 102/ 556977.58 | 18.31 | ref | |
| Urban | 30/ 145757.72 | 20.58 | 1.17 (0.78-1.76) | 0.45 |
| **Socioeconomic Status** | | | | |
| First tertile | 49/ 245517.87 | 19.96 | ref | |
| Second tertile | 41/ 217270.04 | 18.87 | 0.94 (0.62-1.42) | 0.759 |
| Third tertile | 42/ 239947.39 | 17.5 | 0.87 (0.58-1.32) | 0.524 |
| **Body Mass Index** | | | | |
| Normal | 72/ 247796.18 | 29.06 | ref | ref |
| Underweight | 13/30711.69 | 42.33 | 1.45 (0.80-2.62) | 0.216 |
| Overweight and Obese | 47/424149.66 | 11.08 | 0.38 (0.26-0.55) | **0.000** |
| **Physical activity** | | | | |
| Inactive | 92/330810.59 | 27.8 | ref | |
| Active | 38/358413.03 | 10.6 | 0.38 (0.26-0.56) | **0.000** |
| **Opiate ever used** | | | | |
| No | 74/594559.89 | 12.45 | ref | **0.000** |
| Yes | 58/108175.41 | 53.62 | 4.28 (3.04-6.04) | |
| **Cigarette ever used** | | | | |
| No | 61/588239.99 | 10.37 | ref | |
| Yes | 71/114495.31 | 62.01 | 5.98 (4.25-8.42) | 0.000 |
| **Animal contact** | | | | |
| Never | 9/48551.29 | 18.54 | ref | |
| Ever | 123/654184.01 | 18.8 | 1.011 (0.51-1.99) | 0.974 |
| **Source of energy for any purpose** | | | | |
| Gas | 107/608479.22 | 17.58 | ref | |
| Other | 22/86410.42 | 25.46 | 1.49 (0.94-2.35) | **0.090** |

* The Hazard Ratio for age is presented for each year increase in age.

 

**Table 2. Multiple and sensitivity analyses of the impact of pre-diagnostic factors on lung cancer incidence.**

| Multiple analysis | | |
|---|---|---|
| | HR | P value |
| **Gender** | | |
| Male | 1.67 (1.08-2.59) | **0.021** |
| Female | ref | |
| **Age*** | 1.04 (1.02-1.06) | **0.000** |
| **BMI** | | |
| Underweight | 0.98 (0.51-1.85) | 0.939 |
| Overweight and obese | 0.55 (0.37-0.81) | **0.003** |
| Normal | | |
| **Physical activity** | | |
| Active | 0.71 (0.46-1.07) | 0.104 |
| Inactive | ref | |
| **Opiate ever used** | | |
| Yes | 2.20 (1.49-3.26) | **<0.001** |
| No | ref | |
| **Cigarette** | | |
| total pack year | 1.02 (1.01-1.02) | **<0.001** |
| **Source of energy for any purpose** | | |
| Other | 1.73 (1.09-2.76) | **0.020** |
| Gas | ref | |
| **Sensitivity analysis** | **HR** | **P value** |
| **Gender** | | |
| Male | 1.65 (1.03-2.64) | **0.036** |
| **Age** | 1.04 (1.02-1.06) | **0.000** |
| **BMI** | | |
| Underweight | 1.06 (0.54-2.08) | 0.865 |
| Overweight and obese | 0.46 (0.30-0.71) | **0.000** |
| **Physical activity** | | |
| Active | 0.62 (0.39-0.98) | **0.044** |
| **Opiate ever used** | | |
| Yes | 1.87 (1.22-2.87) | **0.004** |
| **Cigarette** | | |
| Total pack year | 1.02 (1.01-1.02) | **0.000** |
| **Source of energy for any purpose** **Other** | | |
| Gas (ref) | 1.78 (1.08-2.95) | **0.024** |

The sensitivity analysis, which excluded patients diagnosed with lung cancer within the first two years of follow-up, considered 108 cases. The analysis revealed age (HR: 1.04 per year, 95% CI: 1.02–1.06), male sex (HR: 1.65, 1.03–2.64), history of opium use (HR: 1.87, 1.22–2.87), total pack-years of cigarette smoking (HR: 1.02 per pack-year, 1.01–1.02), and the use of non-gas energy sources for heating or cooking (HR: 1.78, 1.08–2.95) as risk factors for lung cancer. Conversely, an active lifestyle (HR: 0.62, 0.39–0.98) and a BMI of 25 or higher (HR: 0.46, 0.30–0.71) were negatively correlated with the occurrence of lung cancer.

## Discussion

In our analysis, the age and sex-adjusted incidence rate of lung cancer was determined to be 18.78 cases per 100,000 person-years, which breaks down to 32.19 for males and 9.42 for females. Survival rates one year and five year post-diagnosis were notably low at 18.7% and 1.6%, respectively. Our sensitivity analysis revealed that advanced age, male sex, a history of opiate and cigarette use, and the use of non-gas energy sources significantly increased the risk of lung cancer. Conversely, an active lifestyle and a BMI of 25 or higher were associated with a reduced risk of developing lung cancer.

Building on these findings, we examined previous studies to contextualize our results within a broader epidemiological framework. A local study in Golestan province from 2004 to 2016 reported an age-standardized incidence rate (ASIR) of 12.0 per 100,000 person-years. In 2016, this rate was 21.3 for men and 11.4 for Golestan women per 100,000 person-years [15]. In the global context, the ASIR for tracheal, bronchial, and lung cancer in 2019 stood at 27.7 per 100,000 persons, with higher rates observed in males (40.4 per 100,000) compared to females (16.8 per 100,000) [3]. In developed regions, the reported incidence rates were substantially higher. For example, from 2017 to 2021, the United States reported an ASIR of 53.1 per 100,000 individuals for lung and bronchial cancer [21]. Similarly, in the United Kingdom from 2016 to 2018, Scotland recorded an ASIR of 102.2 per 100,000 and Northern Ireland 84.2 per 100,000, with an overall average of 79.0 per 100,000 [22].

As we analyze the variations in incidence rates across different regions, it becomes crucial to understand how these differences reflect on survival rates. Comparable to our findings, a study conducted in West Azerbaijan, Iran, during 2007–2014 revealed one, two, and three-year survival rates for lung cancer at 39%, 18%, and 0.07%, respectively[23]. In comparison, developed regions reported higher survival rates. The United States reported a five-year lung cancer survival rate of 19% between 2008–2014, which improved to 23% during 2012–2018 [24, 25]. In the United Kingdom, during 2013–2017 the age-standardized one and five-year survival rates for lung cancer for both genders was 40.6% and 16.2%, respectively [26]. Findings from China, within a multicenter cohort study for patients diagnosed between 2011 and 2013, indicated 5-year overall and lung cancer-specific survival rates of 37.0% and 41.6%, respectively [27]. Our lower survival rates imply that our health care for incident lung cancer patients may not be adequate.

Thus, although the reported incidence of lung cancer in this study is comparatively low compared to that of developed countries, the disproportionately lower survival rates suggest that many cases may not be detected until they have reached advanced stages. This likely reflects an underestimation of the true lung cancer incidence, consistent with findings from resource-constrained settings. For instance, How et al. reported that in Pahang, Malaysia, the majority of lung cancer cases were detected at advanced stages, with a median survival of only 18 weeks. Notably, about 15% of their patients survived beyond one year, reflecting survival outcomes similar to those observed in our study [13].

The importance of early detection in improving lung cancer prognosis is well-established. An illustrative example can be found in a Taiwanese study that reviewed lung cancer survival in light of the country's public health strategies. Initiatives such as anti-smoking measures, starting in 1997, the introduction of tyrosine kinase inhibitors in 2004, and the commencement of a low-dose computed tomography (CT) screening trial in 2015, have collectively played significant roles in improving lung cancer outcomes. These efforts have notably enhanced the early detection of lung cancer, contributing to an increase in reported incidence rates. Concurrently, there has been a significant reduction in mortality rates. Data from the National Taiwan University Hospital demonstrates the five-year survival rate for lung cancer markedly improved from 22.1% during 2006–2011 to 54.9% during 2015–2020. These findings underscore the detrimental effects of delayed diagnoses on lung cancer outcomes and highlight the effectiveness of comprehensive public health policies in improving survival rates [27]. While our study is limited by the absence of detailed staging data, which prevents definitive conclusions, the potential impact of late-stage diagnosis on the observed lower survival rates warrants consideration. This issue is especially critical as many participants reside in rural areas where healthcare facilities may lack advanced diagnostic tools. The absence of an established lung cancer screening program in our population likely elevates the possibility of

detecting cases at later stages. Moreover, the economic constraints within our country may influence the treatment pathways available to patients, contrasting with those in higher-income countries, where access to advanced diagnostics and treatments is often linked with better survival outcomes in lung cancer [28]. These findings highlight the critical need for policymakers to reform healthcare services, especially in under-resourced areas. It is essential to develop and implement a comprehensive strategy that identifies individuals at elevated risk of lung cancer. Concurrently, increasing public awareness about the early symptoms of lung cancer is vital. Proactive measures in these domains are essential to mitigate the consequences of delayed diagnoses and to advance the goal of equitable healthcare access and improved patient outcomes across all communities.

In addressing the necessity for comprehensive health reforms, our analysis delves into the specific environmental and lifestyle factors that influence the onset of lung cancer. This focus aims to better tailor public health initiatives and preventive strategies. Smoking, universally acknowledged as the primary risk factor for lung cancer, emerged as a significant variable in our study cohort. Our quantitative findings reveal that each pack-year of smoking is associated with a 2% increase in lung cancer risk. Given that approximately 85% of lung cancer cases are attributable to smoking, the implementation of robust smoking control policies remains the most critical preventive measure against lung cancer [5].

Household air pollution, primarily from burning solid fuels, is a known risk factor for cancers of the trachea, bronchus, and lung [3]. Since a significant portion of time is spent indoors, the quality of the indoor environment greatly influences exposure to these pollutants. The utilization of 'dirty' fuels—such as coal, kerosene, and various forms of biomass, including wood, charcoal, animal dung, and crop waste—markedly contributes to household air pollution and poses a substantial challenge to global environmental health [29]. Our study found that using non-gas energy sources was associated with an increased risk of lung cancer (HR: 1.78, 1.08–2.95). These findings highlight the urgent need for interventions aimed at reducing reliance on solid fuels and improving indoor air quality as integral components of a comprehensive strategy to prevent lung cancer.

In our study, a significant association was observed between opium use and an increased risk of lung cancer (HR: 1.87, 1.22–2.87). This finding aligns with prior research conducted within the GCS, which demonstrated a dose-dependent relationship between opium consumption and the incidence of various cancers, including lung cancer [30]. Further supporting these findings, two recent systematic reviews and meta-analyses have confirmed a strong link between opium use and an elevated risk of various cancers, particularly respiratory cancers. However, further research is imperative to clarify the specific pathophysiological mechanisms underlying this association [31, 32]. Given these insights, there is a critical need for the development of global policies aimed at reducing opium misuse and addressing the severe long-term health consequences of opiate addiction.

Variations in lung cancer incidence also reflect underlying sex disparities. Biologically, sex differences arise from the distinct genotypes of males and females, while gender differences are shaped by societal roles and behaviors linked to gender identity [33]. Our study found a higher risk of lung cancer in males (HR: 1.65, 1.03–2.64), which persisted even after adjusting for smoking and opium use. Determining whether this risk is attributable to male-specific biological factors or confounders such as occupational exposure requires further investigation.

Smoking rates have historically been higher among men, but the gap has been narrowing as smoking becomes more prevalent among women [34]. However, cultural factors in our study region have maintained a low prevalence of smoking among women, with only 1.6% of females reporting smoking habits versus 38.6% of males.

Recently, the changing landscape of lung cancer in women has become a focal point of research. Between 1990 and 2019, while the age-standardized incidence rate of lung cancer consistently decreased in males, it conversely increased among females, despite being lower overall compared to males [3]. Despite the rising trend of lung cancer in women in many regions, primarily linked to the increased female smoking, evidence indicates a higher incidence of lung cancer in never-smoking females in comparison with males. Additionally, some studies show that females are diagnosed with lung cancer at younger ages and with lower levels of tobacco exposure than males, highlighting the

influence of non-smoking related factors such as environmental carcinogens, genetic susceptibility, and hormonal influences [33, 35]. Currently, lung cancer screening criteria place strong emphasis on age and tobacco use. As a result, many women who are at risk—particularly those who smoke less or not at all—may not meet standard screening requirements, potentially delaying diagnoses [33]. These insights underscore the need for nuanced research into sex and gender disparities in lung cancer to enhance our understanding and improve screening and prevention strategies.

In our research, we observed an inverse association between BMI and lung cancer incidence, aligning with multiple studies that have reported a lower risk of lung cancer among individuals with higher BMI [36, 37]. However, the complexity of this relationship is highlighted by the inconsistent results from various studies. Some of these have employed Mendelian randomization analyses, an approach designed to mitigate confounding factors and reverse causation issues, to further investigate this association [38, 39]. For instance, Jiang et al. analyzed a large Norwegian cohort and initially observed an inverse relationship between BMI and lung adenocarcinoma. However, using Mendelian randomization, they determined that this association might be attributable to confounding factors rather than a true causal relationship [40] Given the diverse research outcomes, we positioned our findings within the context of systematic reviews. The 2019 meta-analysis by Gao et al., encompassing twenty-eight prospective cohort studies, showed a relative risk (RR) of 0.77 for the highest BMI category compared to the normal range (95% CI: 0.72-0.82). This inverse relationship, however, was not significant when stratified by smoking status or cancer histopathology, becoming non-significant among never-smokers and for those with small cell lung carcinoma. Further analysis that adjusted for time delays and preexisting weight loss indicated no significant negative correlation (RR: 0.89; 0.66-1.19).[36]. Nevertheless, our study maintained the inverse correlation after excluding lung cancer diagnoses made in the first two years of follow-up. Systematic reviews by Duan et al. in 2015 and Yang et al. in 2013 have found similar negative associations, though these were diminished or became non-significant when focusing on non-smokers [41, 42].

Skepticism about the inverse BMI-lung cancer association points to the complex interplay between smoking patterns and BMI, other confounding factors, reverse causality, and obesity-related mortality risks [43, 44]. Such factors, while adding to the complexity, do not fully explain the paradox. On the contrary, potential mechanisms, such as the enhanced expression of the p53 tumor suppressor gene in obese individuals, might shed light on the protective association seen in some studies. It is also crucial to consider other indicators of obesity, such as waist circumference (WC) and waist-to-hip ratio (WHR), which some studies suggest may independently increase the risk of lung cancer, separate from BMI [45]. Gao et al. reported an increased lung cancer risk with increase in waist circumferences (RR: 1.26; 1.14-1.39) [36]. The relationship between BMI and lung cancer is undoubtedly complex, warranting detailed research to decipher the intricate mechanisms that underpin this association.

Our research suggests an association between an active lifestyle and a lower risk of lung cancer development (HR: 0.71, 0.39–0.98). It's important to acknowledge that the relationship between physical activity and lung cancer incidence remains a topic of ongoing debate [14]. A 2019 systematic review by Liu et al. found a general negative correlation between physical activity and lung cancer risk, with a combined relative risk (RR) of 0.83 (0.77, 0.90), despite substantial heterogeneity across studies. Yet, this correlation was not statistically significant among non-smokers when stratifying the analysis by smoking status [46]. Conversely, a 2020 review by Rana et al. identified an increased lung cancer risk among men engaged in high levels of occupational physical activity [47]. These contrasting findings highlight the complexity of the concern and underscore the need for more rigorous research. Future studies should be expansive, employing large-scale interventional and prospective designs, and should thoroughly assess domain-specific physical activity, incorporate detailed smoking histories, and account for various intensity levels and occupational exposures. Such detailed and contextual research is crucial to discern the true nature of the relationship between physical activity and lung cancer risk.[47].

## Strengths and limitations

This extensive population-based cohort study boasts an impressive follow-up rate of 99%, effectively addresses potential biases associated with missing data. Our rigorous follow-up and data collection procedures have facilitated a thorough examination of cancer patterns and mortality contributors. The study's robustness is further reinforced by replicating measurements within a sizable subgroup and carefully excluding early-onset lung cancer cases to minimize reverse causation concerns. Rigorous case verification and a focus on histologically confirmed cases underscore our commitment to data accuracy.

Our study, while extensive, is not without its limitations. Notably, we did not gather detailed pathological classifications or cancer staging information, which confines our insight to overall survival times. Additionally, our dataset also lacks a systematic record of changes in exposure over time, leaving out risk factors such as secondhand smoke, occupational carcinogens, and family medical history. Despite adjusting for known confounders and performing sensitivity analyses, we acknowledge the possibility of residual confounding. It is noteworthy, however, that certain risk factors, like e-cigarette use and high levels of outdoor air pollution, were not relevant to our cohort based in Northern Iran. While observational studies like ours are prone to measurement errors, our prospective approach likely reduces the impact of such errors on the outcome.

## Conclusions

Our study highlights the significant burden of lung cancer in resource-limited areas, where survival rates are markedly lower compared to developed regions. Key risk factors identified include extensive smoking history, advanced age, male sex, opiate use, and reliance on non-gas energy sources, while higher BMI and an active lifestyle were associated with a reduced risk of developing lung cancer. These findings underscore the urgent need for public health interventions to mitigate these risk factors and promote protective behaviors. Furthermore, certain observed relationships, such as the obesity paradox, require further investigation to fully understand the underlying mechanisms and to determine whether they represent true and causal relationships. Additionally, improving access to advanced diagnostic and treatment facilities is essential to enhance survival outcomes. Our results call for targeted policies and comprehensive strategies to support underserved populations, ensuring equitable healthcare access and improving lung cancer outcomes across all communities.

## Acknowledgements

We extend our deepest appreciation to all participants of the GCS for their commitment and collaboration throughout the duration of our research. Our sincere thanks go to the GCS Center staff, colleagues within the local health networks, and the healthcare providers in our study area for their invaluable contributions. This work was a joint effort supported by collaborations with the Digestive Disease Research Center at Tehran University of Medical Sciences (Principal Investigator: R.M.), the International Agency for Research on Cancer (Principal Investigator: P.B.), and the National Cancer Institute (Principal Investigator: S.M.D.) and Golestan University of Medical Sciences, Gorgan, Iran.

## Author contributions

**Conceptualization:** SeyedehFatemeh Mousavi, Negar Rezaei, Farzad Pourghazi, Maryam Sharafkhah, Maysa Eslami, Akram Pourshams, Gholamreza Roshandel, Sadaf G. Sepanlou, Reza Malekzadeh.

**Data curation:** Sahar Masoudi, Maryam Sharafkhah, Gholamreza Roshandel, Sadaf G. Sepanlou.

**Formal analysis:** Sahar Masoudi, Maryam Sharafkhah.

**Investigation:** Maryam Sharafkhah, Maysa Eslami.

**Methodology:** Sahar Masoudi, Maryam Sharafkhah, Reza Malekzadeh.

**Project administration:** Reza Malekzadeh.

**Supervision:** Negar Rezaei, Akram Pourshams, Hossein Poustchi, Gholamreza Roshandel, Rasoul Aliannejad, Sadaf G. Sepanlou, Reza Malekzadeh.

**Validation:** Negar Rezaei, Hossein Poustchi, Rasoul Aliannejad.

**Writing – original draft:** SeyedehFatemeh Mousavi, Negar Rezaei, Farzad Pourghazi, Maysa Eslami, Hossein Poustchi, Sadaf G. Sepanlou.

**Writing – review & editing:** SeyedehFatemeh Mousavi, Rasoul Aliannejad, Sadaf G. Sepanlou, Reza Malekzadeh.

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
