## [Decision Letter · Decision Letter 0]

28 Nov 2024

PONE-D-24-35243Survival Assessment and Pre-Diagnostic Risk Factors for Lung Cancer Incidence: Insights from the Golestan Cohort StudyPLOS ONE

Dear Dr. Sepanlou,

Thank you for submitting your manuscript to PLOS ONE. After careful consideration, we feel that it has merit but does not fully meet PLOS ONE’s publication criteria as it currently stands. Therefore, we invite you to submit a revised version of the manuscript that addresses the points raised during the review process.

We look forward to receiving your revised manuscript.

Kind regards,

Sohrab Amiri, PhD

Academic Editor

PLOS ONE

2. In the online submission form you indicate that your data is not available for proprietary reasons and have provided a contact point for accessing this data. Please note that your current contact point is a co-author on this manuscript. According to our Data Policy, the contact point must not be an author on the manuscript and must be an institutional contact, ideally not an individual. Please revise your data statement to a non-author institutional point of contact, such as a data access or ethics committee, and send this to us via return email. Please also include contact information for the third party organization, and please include the full citation of where the data can be found.

Reviewers' comments:

Reviewer's Responses to Questions

**Comments to the Author**

1. Is the manuscript technically sound, and do the data support the conclusions?

Reviewer #1: Yes

Reviewer #2: Yes

2. Has the statistical analysis been performed appropriately and rigorously? 

Reviewer #1: I Don't Know

Reviewer #2: Yes

3. Have the authors made all data underlying the findings in their manuscript fully available?

Reviewer #1: Yes

Reviewer #2: Yes

4. Is the manuscript presented in an intelligible fashion and written in standard English?

Reviewer #1: Yes

Reviewer #2: Yes

5. Review Comments to the Author

Reviewer #1: 1. In the introduction section, add a statement about the importance of early detection of lung cancer.

2. State the inclusion and exclusion criteria for the study.

3. State the code of ethics for the study.

4. Remove duplicate results from the discussion section.

5. Referencing is required in various parts of the discussion. For example, "This finding aligns with prior research conducted within the GCS" requires a reference.

6. Has the study considered that exposure to cigarette smoke in non-smokers can have more severe effects on their health? It would be better if exposure to cigarette smoke in non-smokers was also examined.

7. Rewrite the conclusion.

8. Needs grammatical editing.

Reviewer #2: This manuscript evaluates the impact of survival assessment and pre-diagnostic risk factors on lung cancer.

The study is well-written, clearly addressed, and concluded insights and limitations.

Can you please deliberately revise manuscript for any punctuation or grammar issues?.

6. PLOS authors have the option to publish the peer review history of their article (what does this mean? ). If published, this will include your full peer review and any attached files.

**Do you want your identity to be public for this peer review?** For information about this choice, including consent withdrawal, please see our Privacy Policy .

Reviewer #1: No

Reviewer #2: No

---

## [Author Response · Author response to Decision Letter 0]

7 Jan 2025

Dear Prof. Amiri,

We really appreciate that you are reconsidering our manuscript entitled “Survival Assessment and Pre-Diagnostic Risk Factors for Lung Cancer Incidence: Insights from the Golestan Cohort Study” for publication in PLOS One. We would like to thank the editors and reviewers for careful and thorough review of this manuscript, which helped us to improve the quality of this manuscript. The manuscript has been revised for better readability according to the suggestions of the reviewers and editors. We hope it reaches your standards for publication in the journal.

This cover letter includes a point-by-point response to the comments. The changes are also highlighted in the main manuscript.

Sincerely yours,

Sadaf G. Sepanlou

and

Response: We revised the entire manuscript according to PLOS ONE’s style requirements.

2. In the online submission form you indicate that your data is not available for proprietary reasons and have provided a contact point for accessing this data. Please note that your current contact point is a co-author on this manuscript. According to our Data Policy, the contact point must not be an author on the manuscript and must be an institutional contact, ideally not an individual. Please revise your data statement to a non-author institutional point of contact, such as a data access or ethics committee, and send this to us via return email. Please also include contact information for the third party organization, and please include the full citation of where the data can be found.

Response: Thank you for addressing the data availability statement. We designate the Digestive Diseases Research Institute (DDRI) at Tehran University of Medical Sciences as the institutional contact point. The updated data statement is as follows:

Data Availability Statement:

Data are available from the Digestive Diseases Research Institute, Tehran University of Medical Sciences, Tehran, Iran.

Address: North Kargar Street, Shariati Hospital, Digestive Diseases Research Institute.

Phone: 82415000

Fax: 82415400

Email: shariati.ddri@gmail.com

Postal Code: 1411713135

Response: Thank you for your comment. The ethics statement was moved from the end of the manuscript to the methods section.

Response: Thank you for your point. We haven’t cited any retracted article. References were revised according to the requirements of the journal. Per the comment of the respected reviewer,

References 1 and 2 were corrected. Reference 4 was dropped. References 5 and 8 were added to the list. Reference number 21 (20 in former version) was corrected. References number 34 and 37 and 44 were added.

Reviewer(s)' Comments to Author (if any):

Response to Reviewer #1

Comment 1: "In the introduction section, add a statement about the importance of early detection of lung cancer."

Response: Thank you for the insightful recommendation. We have updated the introduction to emphasize the significance of early detection in reducing lung cancer mortality on lines 62-66.

Comment 2: "State the inclusion and exclusion criteria for the study."

Response: Thank you for the suggestion. We have clarified the inclusion and exclusion criteria in the methods section. This statement was added to the methods section: “The exclusion criteria in the original Golestan Cohort Study were unwillingness to participate at any stage of the study for any reason; being a temporary resident; and having a current or previous diagnosis of an upper gastrointestinal cancer.” Specifically, we included all participants diagnosed with lung, tracheal, or bronchial cancers from the beginning of the study and monitored them until their death. While we did not apply specific exclusion criteria, to mitigate potential bias from reverse causation, we conducted a sensitivity analysis excluding cases detected within the first two years of follow-up.

Comment 3: "State the code of ethics for the study."

Response: Thank you for highlighting the importance of clearly stating the ethical considerations of our study. We have ensured that the ethics statement is comprehensively included in the Methods section of our manuscript. This statement details the ethical approvals obtained from the Digestive Diseases Research Institute of the Tehran University of Medical Sciences (Ref: FWA00001331), the U.S. National Cancer Institute (NCI), and the International Agency for Research on Cancer (IARC) (Ref: CN/23/3). Additionally, we have confirmed that all participants provided written informed consent prior to their participation in the study.

Comment 4: "Remove duplicate results from the discussion section."

Response: Thank you for your valuable feedback. We have carefully revised sections of the discussion to eliminate any duplication of results.

Comment 5: "Referencing is required in various parts of the discussion. For example, 'This finding aligns with prior research conducted within the GCS' requires a reference."

Response: Thank you for highlighting the importance of proper referencing. While the sentence mentioned already includes an appropriate citation, we have thoroughly reviewed the entire discussion section to ensure that all statements are adequately supported by relevant references.

Comment 6: "Has the study considered that exposure to cigarette smoke in non-smokers can have more severe effects on their health? It would be better if exposure to cigarette smoke in non-smokers was also examined."

Response: Thank you for raising this important point. While our study did not specifically assess the effects of cigarette smoke exposure in non-smokers, we have acknowledged this as a limitation in the manuscript. We have included a statement in the limitations section to highlight that future research should consider examining the impact of secondhand smoke exposure on health outcomes.

Comment 7: "Rewrite the conclusion."

Response: We appreciate your suggestion to enhance the conclusion of our manuscript. The conclusion has been thoroughly revised to succinctly summarize the key findings of our study, emphasize their significance in the context of lung cancer research, and suggest potential directions for future investigations.

Comment 8: "Needs grammatical editing."

Response: Thank you for pointing out the need for grammatical improvements. We have conducted a comprehensive grammatical review of the entire manuscript to ensure clarity, coherence, and readability.

Response to Reviewer #2

Comment: "This manuscript evaluates the impact of survival assessment and pre-diagnostic risk factors on lung cancer. The study is well-written, clearly addressed, and concluded insights and limitations.

Can you please deliberately revise manuscript for any punctuation or grammar issues?"

Response: Thank you very much for your kind and encouraging feedback on our manuscript. We truly appreciate your positive remarks regarding the clarity and presentation of our study. In response to your suggestion, we have carefully reviewed the manuscript for any punctuation or grammatical issues and made the necessary corrections to enhance the overall readability and quality of the text.

---

## [Editor Report · Decision Letter 1]

27 Feb 2025

Survival assessment and pre-diagnostic risk factors for lung cancer incidence: insights from the Golestan Cohort Study

PONE-D-24-35243R1

Dear Dr. Sepanlou,

We’re pleased to inform you that your manuscript has been judged scientifically suitable for publication and will be formally accepted for publication once it meets all outstanding technical requirements.

Kind regards,

Sohrab Amiri, PhD

Academic Editor

PLOS ONE
---

## [Editor Report · Acceptance letter]

PONE-D-24-35243R1

PLOS ONE

Dear Dr. Sepanlou,

I'm pleased to inform you that your manuscript has been deemed suitable for publication in PLOS ONE. Congratulations! Your manuscript is now being handed over to our production team.

Kind regards,

on behalf of

Dr Sohrab Amiri

Academic Editor

PLOS ONE